# Mitigation of Gaseous Emissions from Swine Manure with the Surficial Application of Biochars

**Zhanibek Meiirkhanuly [1], Jacek A. Koziel [1,*], Baitong Chen [1], Andrzej Białowiec [1,2],
Myeongseong Lee [1,3], Jisoo Wi [1,3], Chumki Banik [1], Robert C. Brown [4,5]
and Santanu Bakshi [5]**

[1]  Department of Agricultural and Biosystems Engineering, Iowa State University, Ames, IA 50011, USA;
    zhanibek@iastate.edu (Z.M.); baitongc@iastate.edu (B.C.); andrzej.bialowiec@upwr.edu.pl (A.B.);
    leefame@iastate.edu (M.L.); jswi@cnu.ac.kr (J.W.); cbanik@iastate.edu (C.B.)
[2]  Faculty of Life Sciences and Technology, Wroclaw University of Environmental and Life Sciences,
    37a Chełmonskiego Str., 51-630 Wrocław, Poland
[3]  Department of Animal Biosystems Science, Chungnam National University, Daejon 34134, Korea
[4]  Department of Mechanical Engineering, Iowa State University, Ames, IA 50011, USA; rcbrown3@iastate.edu
[5]  Bioeconomy Institute, Iowa State University, Ames, IA 50011, USA; sbakshi@iastate.edu
*   Correspondence: koziel@iastate.edu; Tel.: +1-515-294-4206

**Abstract:** Environmental impact associated with odor and gaseous emissions from animal manure is one of the challenges for communities, farmers, and regulatory agencies. Microbe-based manure additives treatments are marketed and used by farmers for mitigation of emissions. However, their performance is difficult to assess objectively. Thus, comprehensive, practical, and low-cost treatments are still in demand. We have been advancing such treatments based on physicochemical principles. The objective of this research was to test the effect of the surficial application of a thin layer ($\frac{1}{4}$ inches; 6.3 mm) of biochar on the mitigation of gaseous emissions (as the percent reduction, % $R$) from swine manure. Two types of biochar were tested: highly alkaline and porous (HAP) biochar made from corn stover and red oak (RO), both with different pH and morphology. Three 30-day trials were conducted with a layer of HAP and RO (2.0 & 1.65 kg·m$^{-2}$, respectively) applied on manure surface, and emissions of ammonia (NH$_3$), hydrogen sulfide (H$_2$S), greenhouse gases (GHG), and odorous volatile organic compounds (VOCs) were measured. The manure and biochar type and properties had an impact on the mitigation effect and its duration. RO significantly reduced NH$_3$ (19–39%) and $p$-cresol (66–78%). H$_2$S was mitigated (16~23%), but not significantly for all trials. The phenolic VOCs had relatively high % $R$ in most trials but not significantly for all trials. HAP reduced NH$_3$ (4~21%) and H$_2$S (2~22%), but not significantly for all trials. Significant % $R$ for $p$-cresol (91~97%) and skatole (74~95%) were observed for all trials. The % $R$ for phenol and indole ranged from (60~99%) and (29~94%) but was not significant for all trials. The impact on GHGs, isobutyric acid, and the odor was mixed with some mitigation and generation effects. However, larger-scale experiments are needed to understand how biochar properties and the dose and frequency of application can be optimized to mitigate odor and gaseous emissions from swine manure. The lessons learned can also be applicable to surficial biochar treatment of gaseous emissions from other waste and area sources.

**Keywords:** air quality; air pollution; sustainable animal production; livestock and poultry; waste management; odor; ammonia; hydrogen sulfide; greenhouse gases; volatile organic compounds

## 1. Introduction

Animal production has a significant impact on local and regional air quality because of emissions of volatile organic compounds (VOC), ammonia ($NH_3$) and hydrogen sulfide ($H_2S$), greenhouse gases (GHG), and odor. These emissions have many adverse effects on the environment [1]. Manure management accounts for about 15% of the total GHG emissions from the agriculture sector in the United States [2]. The median emission rates from swine facilities were 2.08 and 0.20 kg·yr$^{-1}$ pig for $NH_3$ and $H_2S$, respectively [3]. Iowa State University Extension and Outreach Air Management Practices Assessment Tool summarized 12 methods of gaseous emissions mitigation in livestock, manure storage, and handling [4]. Farmers use manure additives because of their ease in application and low-cost in comparison with other odor mitigation technologies that can be complex and require expensive equipment. The performance of manure additives is not well studied, especially on a farm-scale. It is estimated that only ~25% of technologies were tested at the farm scale while the remaining mitigation methods and technologies have not emerged from the laboratory or pilot scale [5]. Microbe-based manure additives treatments are the most popular and are marketed and used by farmers for mitigation of emissions. However, their performance is difficult to assess objectively. Most research is concentrated on a particular type of gas that a proposed treatment reduces and sometimes ignore other emissions that can be negatively affected by the treatment [5]. Thus, comprehensive, practical, and low-cost treatments to gaseous emissions are still in demand.

We have been advancing surficial treatments to livestock and poultry manure based on the physicochemical mode of action. These include various mineral sorbents (e.g., zeolites and bentonites), plant-based enzymes (e.g., soybean peroxidase), and urease inhibitors [6–10]. One of the manure additives that could potentially simultaneously reduce different types of gas emission is biochar. Biochar is the carbon-rich material (a.k.a. char, biocoal) obtained from pyrolysis, torrefaction, or gasification of various types of biomass and biowaste [11–16]. Most of the early applications of biochars are focused on soil amendment and nutrient management [17]. Biochar is obtained as the chemical structure of biomass changes when heated in the absence of oxygen (O), resulting in a loss of hydrogen (H), nitrogen (N), and O relative to carbon (C). The C atoms are firmly bound to one another, which is difficult for microorganisms to break down [18]. A considerable number of studies show how biochar positively affects soil parameters [19,20]. The evidence of recent research on applications of biochar in crop and animal production agriculture is reviewed elsewhere [21].

To date, there are only a few research papers published that test biochar application on stored manure to mitigate gaseous emissions. A recent study by Maurer et al., (2017) [22] showed that biochar has the potential for being an economical and effective treatment for odorous gaseous emissions from swine deep-pit barn. In experiments conducted by Maurer et al. [22] and Dougherty et al. [23], biochar was applied surficially to swine and dairy manure, respectively. Maurer et al. reported up to 23% reduction of $NH_3$ emissions from surficially-treated swine manure. Interestingly, $CH_4$ was generated (~25% increase in emissions) [22]. Dougherty et al., (2017) [23] reported that biochar made of Douglas fir bark mixed with wood fiber showed up to 80% reduction of $NH_3$. A summary of current research on the uses of biochar as a manure additive and its effect on gaseous emissions is presented in Table S1 (Supplementary Materials).

The mechanism of biochar in mitigation of gaseous emissions is complex, and while it can reduce some emissions, it can also increase emissions for other gases (e.g., GHGs in [22]). Biochars are known to affect GHGs emissions, particularly evidenced in the research related to soil. Feng et al. [24] reported that biochar application on the soil during the rice-growing season could reduce up to 91% of $CH_4$ emissions. Moreover, Brassard et al. [25] reported $N_2O$ emission reduction after biochar application from 42–92% in a short-term incubation study. According to Brassard et al. [25] and Rogovska et al. [26], biochar application on soil may increase $CO_2$ emission from 8–91%. The summary of the mitigation effects of gaseous emissions from the soil is presented in Table S2 [25–27].

The biochar properties affect the complex physical, chemical and biological mechanisms and could be explored for gaseous emissions mitigation. Biochars have a high sorption capacity that could

be useful for gas filtration. Results from an experiment conducted by Komnitnas et al., (2016) [28] show that biochar can adsorb 77% of phenol and 26% of $NH_3$ [29]. A mini-review of research on the uses of biochar as an adsorbent for different gases in laboratory-scale experiments is summarized in Table S3 [28–31].

Highly alkaline and porous biochar (HAP) is a result of the addition of 95% of $N_2$ and 5% of $O_2$ during a pyrolysis process [32]. HAP has a high pH, which potentially can be effective in mitigating emissions of $H_2S$ and volatile fatty acids (VFAs). Most of the bacteria producing malodor and $H_2S$ have pH in the range of 6.5–7.5 [33]. HAP has high porosity and a large surface compared with conventional process biochar (such as the red oak, RO) of lower pH. Most recently, we reported on the spatial and temporal effects on the pH in the liquid–air interface after the surficial application of HAP and RO biochars to water [34] and swine manure [35].

This research reports on the next logical step, i.e., evaluation of the mitigation effect of biochars on gaseous emissions from swine manure. The objective of this research was the initial evaluation of HAP biochar and common RO biochar treatment of gaseous emissions from swine manure storages. Specifically, we evaluated:

- Mitigation of odor
- Reduction of $H_2S$, $NH_3$, GHG, and VOCs emissions
- How long biochar treatment lasts after application to manure surface.

## 2. Materials and Methods

### 2.1. Manure

Swine manure was collected from three different farms. Larger volumes were collected from deep pit manure storage barn of AG 450 farm of Iowa State University (Trial 1), Iowa Select Farm (ISF) in North Central Iowa (Trial 2), and outdoor storage in Prestage Farms (PF) in Mid-West Iowa (Trial 3). The 2-m long handle with a scoop was used to collect manure from the manure pits, and manure was used for the experiment within a week. The pH of manure samples was measured with the Accumet Refillable electrode [36] and Accumet AB 15 pH meter (Fisher Scientific, Kansas City, KS, USA). Total solids were determined by mass loss after drying [37]. The total volatile solids were derived by muffling samples for 2 h at 550 °C [38]. Total N was estimated by a thermal conductivity cell [37].

At the end of each Trial, four containers of the same treatment were mixed and emptied into a 3 L jar, then kept in a refrigerator at 4 °C for manure analyses. After the end of all three Trials, 12 samples (four per Trial, i.e., each Trial set consisting of $n = 1$ pre-trial, $n = 1$ control, $n = 1$ RO, and $n = 1$ HAP) were sent to Brookside Laboratories, Inc. (New Bremen, OH, USA) for analyses of total solids, total volatile solids, and total N. The properties of the manures are shown in Table 1.

**Table 1.** Properties of swine manure (pit- or outdoor stored) used in the experiments.

| Properties | AG 450 (Trial 1) (Pit Storage) | ISF (Trial 2) (Pit Storage) | PF (Trial 3) (Outdoor Storage) |
|---|---|---|---|
| pH | 7.47 | 8.00 | 7.55 |
| Total solids (%) | 2.64 | 4.07 | 2.60 |
| Total volatile solids (%) | 66.67 | 71.01 | 66.54 |
| Total Nitrogen (%) | 16.10 | 13.37 | 11.88 |

### 2.2. Biochar Properties

Tested biochars were made from red (RO) oak and corn stover (highly alkaline and porous biochar, HAP). Elemental content (C, H, N, and S) was measured by a C/N combustion analyzer (Vario Microcube, Elementar Analysensysteme GmbH, Langenselbold, Germany) [39]. Thermogravimetric analysis (TGA) on a Mettler-Toledo TGA/DSC STARe System was used with N flow at 100 mL·min$^{-1}$ for the

proximate analysis of biochar. Fixed C content was estimated by subtracting percentages of moisture, volatile matter, and ash content from the sample. Results of the elemental analyses, moisture/fixed C, volatile matter, and pH are summarized elsewhere [34]. In addition, a solid addition method was used to analyze the zero point charge (ZPC) of biochars [40]. Pore images of biochars were taken using the scanning electron microscopy (SEM) method (FEI Quanta 250 FE-SEM) [39,41]. Results of SEM analyses are summarized elsewhere [34]. Fourier transforms infrared (FTIR) analysis was conducted to analyze functional groups in the biochar by using Thermo Scientific Nicolet iS20 (Thermo Fisher Scientific Inc., Waltham, MA, USA) with attached iTR accessory [42].

### 2.3. The Experimental Stand Description

The study of two types of biochar on odor mitigation was completed in lab-scale manure storage simulators ('containers', 'reactors'). Manure (800 mL) was put into 12 1700 mL glass containers (19 cm × 14.5 cm × 7.5 cm). Thus, the 900 mL remained as a controlled headspace. The headspace was continuously flushed with 100 mL·min$^{-1}$ and controlled by rotameters (Dwyer, RMA-11-SSV, Michigan City, IN, USA) with valves to adjust a ventilation rate of seven headspace exchanges per hour to simulate farm-scale ventilation of pit manure storages. Air from the compressed air source was flowing through the hydrocarbon filter (Sigma-Aldrich, St. Louis, MO, USA) and then directed to the mass flow controller (MFC, Aalborg, range of 0–1000 mL·min$^{-1}$, Orangeburg, NY, USA) that controlled total flow. After passing the MFC, the air was flowing to a manifold that distributed the air equally to flush the headspace of manure (Figure 1, Figure S1).

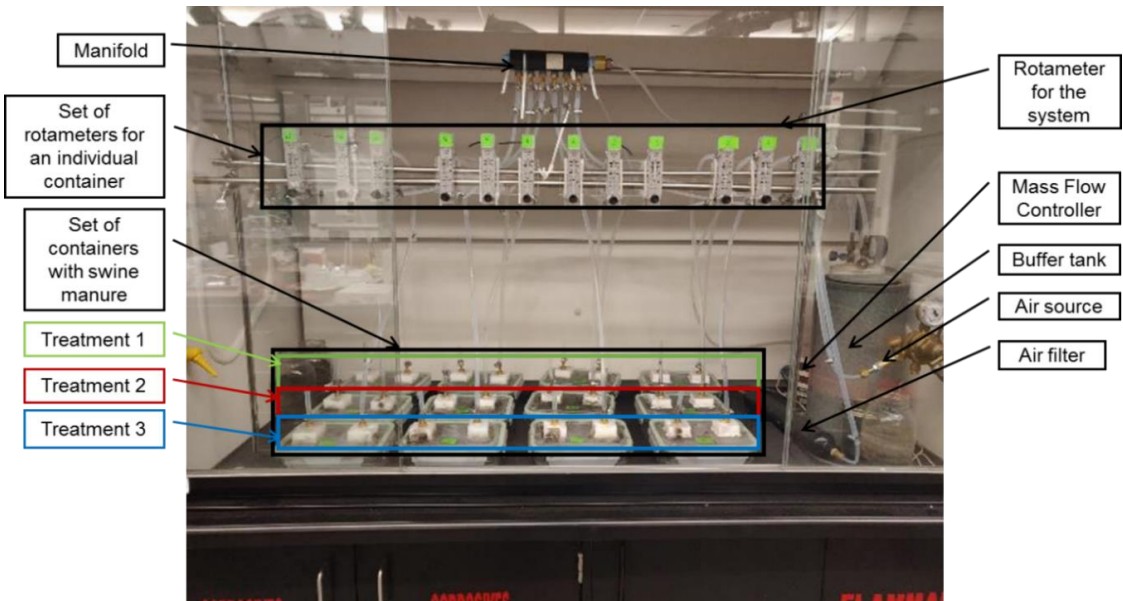

**Figure 1.** A laboratory-scale system to test the effects of surficial biochar application to swine manure on gaseous emissions.

### 2.4. Experimental Design

The laboratory-scale experiment was designed to imitate manure in a deep pit swine manure storage. The experimental design was a completely randomized design for three trials with 30 days of duration and 1 week of stabilization before each Trial (i.e., before application of biochars on Day 1). The dose of 1/4″ (6.35 mm) of the biochar layer applied on manure was selected according to Maurer et al. [22]. Manures from AG 450, ISF, and PF were used in Trials 1, 2, and 3, respectively. Each Trial consisted of testing four reactors of fresh swine control manure, four reactors of fresh swine manure treated with RO biochar at a dosage of 1.65 kg·m$^{-2}$, and four reactors of fresh swine manure treated with HAP biochar at a dosage of 2 kg·m$^{-2}$ (Table 2, Figure S1 in Supplementary Materials).

**Table 2.** Matrix of the experiment.

| Trial | RO Biochar Dose (kg·m$^{-2}$) | HAP Biochar Dose (kg·m$^{-2}$) | Control | Manure Source | Trial Length (d) |
|---|---|---|---|---|---|
| 1 | 1.65 | 2.0 | 0 | AG 450 | 30 |
| 2 | 1.65 | 2.0 | 0 | ISF | 30 |
| 3 | 1.65 | 2.0 | 0 | PF | 30 |

Note: mass of RO per reactor = 40.1 g; mass of HAP per reactor = 48.6 g.

*2.5. Gas Concentration Measurements*

2.5.1. Ammonia and Hydrogen Sulfide

To measure $NH_3$ and $H_2S$ concentrations, the real-time system (OMS-200, Smart Control & Sensing Inc., Daejeon, Korea) attached with electrochemical gas sensors from Membrapor Co. (Wallisellen, Switzerland) [43] was sampling the air for 5 min with a suction rate of 2 L·min$^{-1}$. Due to the restrictions of the laboratory scale deep barn pit simulation system, gas samples were collected from four containers of the same treatment simultaneously into one Tedlar bag. A manifold with four ports, connected to four containers via Teflon tubes, was attached to a vacuum chamber with a suction rate of 350 mL·min$^{-1}$. A Tedlar bag was put in the chamber for 35 min to collect ~12.25 L of the air sample. After the collection of the gas sample, the Tedlar bag was attached to the real-time analyzing system, which measured the concentration of $NH_3$ and $H_2S$ in ppm. Gases from each treatment were collected in and analyzed in triplicates (Figure S2), i.e., three Tedlar bags were filled with gas samples for each measurement, and $NH_3$ & $H_2S$ concentrations were measured separately by OMS-200.

2.5.2. Greenhouse Gases

A gas chromatograph (SRI Instruments, Torrance, CA, USA) with a flame ionization detector (FID) and electron capture detector (ECD) was used to measure GHG concentrations. The GHG analyses followed the processes used earlier [22]. The instrument was calibrated with standard gases in triplicates with 1005 and 4020 ppm $CO_2$, 0.1 and 1.0 ppm $N_2O$, and 10.5 and 20.5 ppm $CH_4$.

A four-port manifold with an attached sampling port was connected to two pocket pumps with a pumping rate of 175 mL·min$^{-1}$ each. After connecting the manifold to a block of four same treatment containers via Teflon tubes, triplicates of samples were collected through a sampling port using a syringe and then injected into the vacuum of 5.9 mL Exetainer (Labco Limited, Wales, UK) vials, which were previously cleaned by seven cycles of He flushing and vacuuming. Then, samples were analyzed with a GC (Figure S3). Resulting concentrations were estimated based on calibration curves developed with standard gases.

2.5.3. Volatile Organic Compounds

Gas samples were also composited for the VOC analyses greatly reducing the variability inherent with gaseous emissions from manure. A four-port manifold was connected to a block of four (identical) treatment containers. A clean glass bulb with a volume of 1 L was connected to the manifold and on the other side to two pocket pumps with a suction rate of 175 mL·min$^{-1}$ to flush air through the glass bulb. After 15 min of flushing, connections ports were closed, and the bulb was detached from the manifold and the pumps. Solid-phase microextraction (SPME) fiber assembly divinylbenzene/Carboxen/polydimethylsiloxane (DVB/Car/PDMS) with a length of 1 cm was inserted into the bulb through septa on it. After 10 min of passive sample collection, the SPME was injected into a thermal desorption-multidimensional GC-MS/olfactometry system (TD-MDGC-MS/O) by Microanalytics (Round Rock, TX, USA) for separation and analysis of VOCs. The details of the instrument are described in Zhang et al. [44]. Targeted compounds such as isovaleric acid, phenol, *p*-cresol, indole, and skatole were scanned with selected ion monitoring (SIM) mode (Figure S4). We initially tracked dimethyl disulfide, dimethyl trisulfide, propanoic acid, isobutyric acid, butyric acid,

and valeric acid. However, they were not consistently detected in all three trials and consequently are not reported here.

## 2.6. Statistical Analysis

This experiment was a completely randomized design; all observations were assumed to be independent, normally distributed, and have a similar variance. The One-Way Analysis of Variance (ANOVA) was performed to compare the means of treatments and control for each trial by using the R program (R i386 3.6.0. Ink). When the calculated *p*-value is less than 0.05, the % *R* of the treatment is statistically significant.

## 2.7. Data Analysis

The effectiveness of biochar treatment on the mitigation of gaseous emissions was evaluated as follows. The overall % reduction for each measured gas and each Trial was estimated as (Equation (1)):

$$\% \ R = \ \frac{(C_{Con} - C_{Treat})}{C_{Con}} \times 100\% \tag{1}$$

where: % *R* = % reduction; $C_{Con}$ = sum of measured concentrations in control on Day 1 to Day (n) divided by the number (n) of days; $C_{Treat}$ = sum of measured concentrations in treatment on Day 1 to Day (n) divided by the number (n) of days.

The % *R* was estimated for each Trial. The average % *R* (Avg % *R*) was estimated based on the average $C_{Con}$ and $C_{Treat}$ from three Trials. The peak area counts (PACs) were used as the surrogate metric for VOC concentrations. Gaseous emissions for $NH_3$, $H_2S$, and GHGs were also tracked by temporal estimates of flux. The flux was estimated as the product of measured $C_{Con}$ & $C_{Treat}$, airflow in the headspace of manure storage simulators (100 mL·min$^{-1}$), divided by emitting manure area (assumed to be equal to 19 cm × 14.5 cm) [22]. The flux is useful for verifying the range of estimated flux on a lab-scale with other published research.

# 3. Results

## 3.1. Observation of Biochar Layer Interaction with Swine Manure

In trial 1, HAP biochar started sinking right after application. At the end of the trial, there was a thin layer of wet biochar floating near the surface; some biochar was in suspension, while most of it settled on the bottom. All RO was wet by day 4. RO had a thicker wet layer floating near the surface of the manure and settled fewer particles on the bottom in comparison with HAP.

During trial 2, RO biochar was floating for the first 2 weeks on top of the manure, and then it started to incorporate into manure after day ~15. On the last day of the trial, about half of RO biochar was wet and was floating near the top of the manure; another half settled on the bottom. HAP biochar absorbed liquid manure where it was interacting with it, then crusted and partially separated from manure. Thus, a headspace between the HAP layer and manure was created near the perimeter, and only ~30% of HAP biochar was still in contact with manure near the center of the storage container (Figure S5).

In trial 3, HAP biochar started to incorporate into manure on day 3 and completely sank 8 days after application. At the end of the trial, all of the HAP biochar settled on the bottom of the container. A layer of RO biochar applied on the manure started getting wet after 2 weeks, and at the end of the trial, half of the RO biochar became wet and was floating on the manure surface while another half settled on the bottom of the container (Figure S6).

Our working hypothesis was that the biochar layer influences manure pH, which could inhibit gas transfer from liquid to air. However, as is shown in Table 3, because of the differences between pre-trial and other treatment manures as well as because there were no significant differences between control and biochar-treated manures over all three trials, no effect of biochar on the manure pH can be

assumed. Biochar-treated manure gained solids and volatile solids in comparison with the control manure. The total N content dropped 1.5–2 times compared to control over all three trials, likely due to the high biochars porosity that might adsorb N (Table 3). The possibility of N adsorption and possible slow release to (e.g.,) soil after biochar-treated manure is applied to the land should be further explored as the mitigation of gaseous emissions could be followed by agronomic benefits to the soils.

**Table 3.** Physical properties of manure after trials, control, RO, & HAP-treated manures.

| Trial | Treatment | pH | Total Solids (%) | Total Volatile Solids (%) | Total Nitrogen (%) |
|-------|-----------|------|------------------|---------------------------|--------------------|
|       | Control   | 9.16 | 2.17             | 48.85                     | 10.65              |
| 1     | RO        | 9    | 8.16             | 84.44                     | 3.13               |
|       | HAP       | 9.03 | 4.87             | 60.78                     | 4.78               |
|       | Control   | 9.07 | 4.08             | 66.18                     | 7.57               |
| 2     | RO        | 8.97 | 10.64            | 85.62                     | 3.32               |
|       | HAP       | 8.99 | 8.81             | 68.56                     | 4.43               |
|       | Control   | 9.07 | 2.41             | 59.75                     | 7.47               |
| 3     | RO        | 8.77 | 4.29             | 63.4                      | 3.89               |
|       | HAP       | 8.75 | 8.33             | 86.19                     | 2.61               |

One of the reasons for different biochars behavior in the meaning of sinking or floating on the manure surface could be a difference in hydrophobicity of biochars. Higher hydrophobicity decreases liquid absorption, which can prevent biochar from sinking in manure [45]. Hydrophobicity of biochar is associated with residual alkyl functional groups, which can be decomposed by a higher temperature of charring. Thus, hydrophobicity depends on feedstock and temperature of charring. It is possible to compare the hydrophobicity of biochars by comparing the peak area in a range of 3000–2800 cm$^{-1}$ on FTIR spectra [46]. As it can be seen in Figure S7, RO biochar has a small peak on that range, while the graph for HAP biochar is flat, which supports the above-mentioned hypothesis. Another property of HAP biochar that made it sink in manure could be high ash content in comparison with RO biochar. In addition, the differences in pore size associated with HAP and RO biochars (visible in SEM images, [34]) could also affect the liquid penetration.

Another parameter that could affect biochar floating on top of manure is the solid content in manure. Trial 1 showed that HAP and RO started to incorporate with manure immediately after application and on day 2, respectively. Then, most of the particles of biochar with the lower solid content completely sank in manure. In trial 2, where manure with higher solid content was used, RO biochar became wet only after ~2 weeks, and there was a floating wet layer of biochar still near the manure surface. In trial 3, manure with total solids higher than in trial 1, RO behaved similarly to trial 2, and HAP settled on the bottom after ~1 week.

### 3.2. Effectiveness of RO and HAP Biochar Treatment of Gaseous Emissions from Stored Swine Manure

The mitigation effects of biochar treatment are summarized in Table 4. Overall, both biochars were effective in mitigating NH$_3$, H$_2$S, and odorous phenolic VOCs. The impact on GHGs, isobutyric acid, and the odor was mixed with some mitigation and generation effects. Clearly, the manure and biochar type and properties had an impact on the mitigation effect and its duration (Tables S4–S14 and Figures S8–S16).

**Table 4.** Effectiveness of RO and HAP biochar treatment of gaseous emissions from stored swine manure. The effect of time (and therefore the duration of the mitigation effect) is reported as the percent reduction (% *R*) of measured concentrations for the first week, first 2 weeks, first 3 weeks, and entire trial, respectively. * The average % *R* (Avg % *R*) was estimated based on the average $C_{Con}$ and $C_{Treat}$ from three trials. ** T1 = trial 1, T2 = trial 2, T3 = trial 3; % *R* was estimated with Equation (1) for each Trial; Bold signifies statistical significance ($p \le 0.05$).

| Gas or Odor | RO Biochar Avg % *R* * (T1/T2/T3) ** | | | | HAP Biochar Avg % *R* * (T1/T2/T3) ** | | | |
|---|---|---|---|---|---|---|---|---|
| | **First week** | **First 2 weeks** | **First 3 weeks** | **Entire Trial** | **First week** | **First 2 weeks** | **First 3 weeks** | **Entire Trial** |
| $NH_3$ | 38 **(41/28/55)** | 30 **(33/20/49)** | 28 **(26/21/43)** | 25 **(19/21/39)** | 30 **(38/29/18)** | 24 **(26/27/**14) | 19 **(22/23/**8) | 16 **(18/21/**4) |
| $H_2S$ | 23 (20/37/9) | 26 **(23/32/23)** | 27 **(22/33/25)** | 20 (23/**19/16**) | 16 (10/29/7) | 20 (8/35/14) | 19 (20/**22**/11) | 16 (22/**19**/2) |
| $CH_4$ | **54** **(51/57/52)** | **33** **(26/37/39)** | −38 (−6/−88/−20) | **−54** (−26/**−104**/−32) | 5 (13/15/**−38**) | **−27** (−30/−8/**−57**) | **−69** (**−35**/−35/**−194**) | **−82** (**−55**/−41/**−221**) |
| $CO_2$ | 16 (0/22/25) | **19** (1/**24**/31) | **16** (0/**20**/29) | 14 (0/**17**/25) | **−31** (**−26**/14/**−114**) | **−19** (−15/17/**−80**) | **−18** (−11/7/**−61**) | **−15** (2/0/−57) |
| $N_2O$ | 8 (6/11/7) | 0 (5/11/−10) | 3 (3/8/−1) | 3 (4/7/−1) | 9 (1/12/12) | 1 (3/8/−6) | 8 (1/9/13) | 23 (−6/3/46) |
| Phenol | 85 (−37/**91/84**) | 84 (8/**85/84**) | 84 (−13/**83/84**) | 84 (−13/**94/84**) | 98 (11/**97/98**) | 98 (25/**96/98**) | 98 (26/**95/99**) | 99 (60/**83/99**) |
| *p*−cresol | **71** (**69**/43/**72**) | 79 (21/**65/80**) | 78 **(77/64/79)** | 78 **(77/66/78)** | 94 **(90/91/95)** | 94 **(89/91/94)** | 93 **(89/93/94)** | 97 **(97/91/97)** |
| Skatole | **60** **(70/60/55)** | **49** **(77/62/**27) | **48** **(72/62/**29) | 47 **(72/66/**29) | 91 **(90/94/90)** | 91 **(88/95/91)** | 90 **(88/89/91)** | 93 **(95/74/93)** |
| Indole | **41** (−68/**69/36**) | 84 (2/**68/88**) | **82** (−13/**74/86**) | 79 (−11/**67/84**) | 84 (89/**90/80**) | 93 (81/**92/94**) | 92 (81/**92/92**) | 73 (94/**92/**29) |
| Isobutyric acid | −23 (−61/−37/19) | 18 (24/−12/**33**) | 17 (8/11/28) | 19 (9/18/**26**) | 31 (32/8/47) | 15 (40/−44/**32**) | 17 (39/−6/**23**) | 61 (88/8/47) |
| Odor | 12 (−13/25/**22**) | 17 (17/15/19) | 23 (**19**/20/32) | **22** (**17**/21/30) | 18 (−1/6/**43**) | 15 (15/17/11) | −2 (**15**/−12/−12) | −30 (−31/−12/−59) |

Note: First week—an average of % *R* for Day 1 & 5; First 2 weeks—an average of % *R* for Day 1, 5, 8, and 12; First 3 weeks—an average of % *R* for Day 1, 5, 8, 12, 15, 19, and 22; Entire trial—an average of % *R* for Day 1, 5, 8, 12, 15, 19, 22, 26, and 29.

In the case of RO treatment, there was a significant percent reduction (% $R$) that ranged from 19~39% for $NH_3$ in all three trials and the types of manure tested. $H_2S$ was mitigated (16~23% reduction), but not significantly for all trials. $CH_4$ was generated (−26 to −104%), but not significantly for all trials. $CO_2$ trended toward mitigation (0–25%), but not significantly for all trials. There was no significant effect on $N_2O$ (−1–7%). Overall, the phenolic VOCs had relatively high % $R$ in most trials. Significant (66–78%) reductions for *p*-cresol were observed for all trials. The % $R$ for skatole ranged from 29–72%, but not significantly for all trials. The % $R$ for phenol and indole ranged from (−13–94%) & (−11–84%) but not significantly for all trials. The % $R$ for isobutyric acid ranged from (9–26%) but not significantly for all trials. Finally, the odor was mitigated in all trials (17–30%) but not significantly for all trials.

The RO treatment resulted in decreasing effectiveness (% $R$) for $NH_3$ mitigation, consistently for the first week, the first 2 weeks, the first 3 weeks, and the entire trial for all three trials (Table 4). The % $R$ for *p*-cresol was relatively steady. However, % $R$ for several targeted gases ($H_2S$, phenol, skatole, indole) were relatively steady in at least one of the trials.

In the case of HAP treatment, the % $R$ for $NH_3$ ranged from 4–21% but not significantly for all trials. $H_2S$ was mitigated (2–22% reduction), but not significantly for all trials. $CH_4$ was generated (−41 to −221%), but not significantly for all trials. $CO_2$ trended toward generation (−57–2%), but not significantly for all trials. There was no significant effect on $N_2O$ (−6–46%). Overall, the phenolic VOCs had relatively high % $R$ in most trials. Significant (91–97%) reductions for *p*-cresol were observed for all trials. The % $R$ for skatole ranged from 74-95% and was significant for all trials. The % $R$ for phenol and indole ranged from (60–99%) & (29–94%) but not significantly for all trials. The % $R$ for isobutyric acid ranged from (8-88%) but not significantly for all trials. Finally, the odor was generated in all trials (−12 to −59%) but not significantly for all trials. It is important to mention that the odor measurement does not capture odor character.

The HAP treatment resulted in decreasing effectiveness (% $R$) for $NH_3$ mitigation, consistently for the first week, the first 2 weeks, the first 3 weeks, and the entire trial for all three trials (Table 4). The % $R$ for *p*-cresol and skatole was relatively steady. However, % $R$ for several targeted gases ($H_2S$, phenol, indole) were relatively steady in at least one of the trials. $CH_4$ generation increased with the duration of most of the trials.

## 4. Discussion

### 4.1. Effect of Biochar Treatment on Ammonia and Hydrogen Sulfide

Biochar addition had a mitigating effect on $NH_3$ emissions. However, the effect was temporal, opening up the possibility of biochar reapplication in subsequent research. On the first day of trial 1, HAP and RO biochars showed as much as 40% ($p < 0.0001$) and 52% ($p < 0.0001$) reduction of $NH_3$, respectively. After that, the reduction rate started to decrease over time (Figure S8, Table S4). The last statistically significant reduction by comparing the same day measurements between the control and treatment was on day 15, where it was 17% ($p = 0.04$) and 27% ($p < 0.0001$) for HAP and RO biochar, respectively (Figure S8). During trial 2, HAP and RO started with 34% ($p < 0.0001$) and 33% ($p < 0.0001$) reduction of $NH_3$ emissions on day 1 and then gradually decreased to 15% ($p = 0.01$) and 12% ($p = 0.04$) on day 26. In trial 3, HAP biochar reduced 41.8% ($p < 0.0001$) of $NH_3$ emission. However, from day 5, it showed no $NH_3$ reduction; in fact, it increased $NH_3$ emissions on some days. RO biochar started with 70.4% of $NH_3$ reduction ($p < 0.0001$) on day 1, and then the reduction rate gradually decreased to 22.2% on day 29 ($p < 0.0001$).

Even though manure pH changed for all treatments, there were no significant differences in pH between the control and biochar-treated manures after each trial, so it cannot be stated that biochar inhibited $NH_4^+$ from transformation to $NH_3$ during the entire Trial. ZPC of biochar could be a parameter that was responsible for $NH_3$ reduction. Because ZPC of biochars was below biochars pH,

their surface was mostly negatively charged, which was favorable to attract $NH_4^+$ cations and keep them on the biochar surface.

During all three trials, manure pH changed from 7.47–8 to 8.75–9.07, and this increase in pH forced more $NH_4^+$ ions into gaseous $NH_3$ form. Thus, $NH_3$ emissions reduction gradually decreased from day 1 to day 29. Interestingly, Takaya et al. reported that the oak biochar has the $NH_3$ sorption capacity of 4–8 mg·g$^{-1}$ of biochar [47]. Assuming this sorption capacity, the RO biochar would be fully saturated by day 7–14. Clearly, the RO biochar in this research was effective longer, e.g., still showing a significant reduction on day 29 in trial 3. Thus, the most probable explanation is that biochar created a semi-porous physical barrier that slowed mass transfer from manure to headspace.

Emissions of $H_2S$ were more challenging to mitigate with biochars, with some interesting temporal effects (Figure S9, Table S5). Trial 1 showed no statistically significant reduction of $H_2S$ due to the effect of RO. Similarly, HAP treatment was not significant, even though it had 100% ($p = 0.07$) of reduction starting on day 19 until the end of the Trial. In trial 2, HAP biochar significantly reduced emissions by 24% ($p = 0.001$) and 39% ($p < 0.0001$) on day 1 and 8, respectively. RO biochar was reducing $H_2S$ emission for the first ~12 days, where the highest reduction was 49% ($p < 0.0001$) on day 5. During trial 3, HAP biochar showed a significant reduction of 25.4% ($p < 0.0001$) till day 8. RO biochar had a significant reduction of 26.2% ($p < 0.0001$) until day 22. However, on day 15, even though RO reduced 43.7% of $H_2S$, it was not statistically significant ($p = 0.6221$).

Shang et al. claimed that tested biochar had a range of sorption capacity from 1.2–121.4 mg·g$^{-1}$, and the most conducive pH for $H_2S$ sorption was the pH higher than pKa of $H_2S$, which is 7 [48]. Shang et al. also stated that FTIR peaks at 3420 and 1730 cm$^{-1}$, which correlated to OH stretching and COO$^-$ stretching respectively, were crucial for $H_2S$ adsorption [48]. Ayiania et al. studied that the mechanism of $H_2S$ sorption is driven by the size of biochar pores [49]. However, our experiments showed that HAP biochar with pH = 9.2 and larger pores [34] performed less $H_2S$ reduction in comparison with RO over all three trials. It should be mentioned that FTIR spectra (Figure S7) showed that RO had higher peaks in OH and COO$^-$ stretching. Most likely, the complexity of swine manure content and competition between gases were not favorable for reducing $H_2S$ emitted from the manure.

## 4.2. Effects of Biochar Treatment on Greenhouse Gas Emissions

On day 1 of Trial 1, HAP and RO treatment had 31% ($p = 0.0001$) and 77% ($p = 0.0001$) of $CH_4$ reduction, respectively. However, after day 12, both started producing $CH_4$, and the highest production was observed on day 26, where HAP and RO treatment had 415% ($p = 0.0001$) and 153% ($p = 0.0001$) of increase of emissions. During trial 2, on day 1, HAP and RO reduced $CH_4$ emission up to 35% ($p < 0.0001$) and 65% ($p < 0.0001$), respectively. However, from day 15, emissions increased up 101% ($p < 0.0001$) and 35% ($p < 0.0001$), respectively. The highest increase in the $CH_4$ emissions of 176% ($p < 0.0001$) and 288% ($p < 0.0001$) was measured on the last day of the trial for HAP and RO, respectively. On trial 3, $CH_4$ was reduced to 57.8% ($p < 0.0001$) on day 1 due to RO biochar, and the reduction gradually decreased to 20.6% ($p < 0.0001$) on day 11. On day 15, RO biochar started the production of $CH_4$, where the highest production was 301.8% ($p < 0.0001$) on day 19. HAP increased $CH_4$ emission during the whole trial, where the highest increase was 671% ($p < 0.0001$) on day 19 (Figure S10, Table S6).

On the first day of trial 1, HAP biochar increased $CO_2$ emission up to 46% ($p = 0.0001$) (Figure S11, Table S7). Then again, on day 26, HAP and RO increased $CO_2$ emission up to 42% and 39%, respectively, but they were not statistically significant. Overall, the $CO_2$ reduction was not significant, and % reduction fluctuated between 2% and −9% (except for day 1 and 26). In trial 2, both HAP and RO reduced $CO_2$ emissions till day 11 where it was 17% ($p < 0.0001$) and 26% ($p < 0.0001$), respectively.

No significant reduction of $N_2O$ was measured during trial 1. The highest reduction was shown on day 11, where HAP and RO had 9% ($p = 0.2$) and 11% ($p = 0.3$) reduction, respectively. $N_2O$ mitigation was statistically significant for the first 2 weeks, and the highest reductions of 17% ($p < 0.0001$) and 16% ($p < 0.0001$) were observed on day 5 for HAP and RO, respectively on trial 2. In trial 3, both HAP and RO biochars increased $N_2O$ emission up to 28.9% ($p < 0.0001$) on day 8 and decreased the emission to

20.8% ($p < 0.0001$) and 40.5% ($p < 0.0001$), respectively, on day 22. No statistically significant reductions were observed on other days of the trial (Figure S12, Table S8).

Jiang et al. and Shen et al. studied that aeration during composting decreased $CH_4$ emission [50,51], and in addition to that, Agyarko-Mintah et al. stated that biochar improved aeration and methanotrophy of compost [52]. Besides the fact that the headspace of the container with manure was continuously flushed with air, freshly applied biochars might enhance manure surface aeration and so decrease methanogenic activity. However, after biochar started to incorporate into manure, it could be a source of carbon for methanogens for $CH_4$ production. Moreover, due to high porosity, biochar was not able to capture smaller $CH_4$ particles. In agreement with the statement, HAP biochar with larger pores was less effective and increased methane emission after 1–2 weeks. Also, in the case of RO biochar, it started producing $CH_4$ after biochar was incorporated into manure, which can be seen in Figure S10 (part b) where biochar sank after day 15, and high $CH_4$ emission was observed on day 19. The theory was that biochar was adsorbing $CH_4$, and after incorporation into manure, it released captured $CH_4$ out of its pores.

RO biochar with less N content (Table 3) and smaller pores showed better performance in $CO_2$ sorption in comparison with HAP biochar. Zhang et al. revealed that micropores of <0.70 nm were responsible for $CO_2$ uptake [53]. Creamer et al. [54] found that a MgO-activated sugarcane bagasse biochar had a $CO_2$ adsorption capacity of 733 mg g$^{-1}$. In this research, RO treatment trended toward mitigation while the HAP treatment trended toward generation, but not significantly for all trials. There was no significant effect on $N_2O$.

### 4.3. Effect on Volatile Organic Compounds

The average % *R* (Table 4) ranged from 41 to 99% for all phenolic VOCs (Figures S13–S16; Tables S9–S12). Also, both biochars had a statistically significant average % *R* of *p*-cresol & skatole, ranging from 71–97% & 47–93%, respectively (Figures S14 and S15). HAP biochar was consistently mitigating indole (average % *R* ranging from 73–93% (Figure S16 & Table S12).

Zhang et al., reported that VOC sorption was mainly governed by surface area and non-carbonized organic matter; however, none of the VOCs from this experiment were tested [55]. In addition, Shen et al., studied that surface area and pore volume were the main contributors in the sorption of gaseous phenol and toluene by activated char [56]. Moreover, according to Lee et al., phenol adsorption decreased with increasing pH of an aqueous solution [57]. In the case of Trial 3, HAP biochar with larger pores showed a higher % *R* of VOC, especially phenol.

The abundance of volatile fatty acids was significantly lower and that could be a reason for a low % *R* for all three trials for the isobutyric acid. For isobutyric acid, both types of biochar did not statistically reduce the emission from deep-pit manure (Trials 1 & 2, Table S13). However, RO biochar reduced the isobutyric acid emission by 26.2% ($p = 0.0002$), and HAP biochar reduced ~47% ($p = 0.003$) in Trial 3 (Table S13), which treated the manure from outdoor storage instead of a deep pit barn.

Finally, RO biochar showed ~22% reduction ($p = 0.008$) in the average odor concentration of the three trials throughout the 1-month testing period (Table S14). HAP biochar showed an average reduction of 17.5% ($p = 0.019$) in the first week, decreased to 15% ($p = 0.019$) reduction in the second week, and showed generation ($p > 0.05$) of odor concentration in the last 2 weeks (Table S14).

The research is needed to develop and test practical technologies for the mitigation of gaseous emissions from stored swine manure. Clearly, the need is still there as the performance data for marketed products is nearly non-existent [4,5]. In a recent publication, Chen et al., (2020a) reported on a comprehensive study to evaluate gas emissions data from swine manure treated with different commercial additive products [58].

Most recently, Chen et al., (2020b) [59] reported on the use of HAP and RO biochar as being very effective in mitigating acute releases of $H_2S$ from agitated manure on a pilot-scale. Up to ~80% reduction of short-term $H_2S$ emissions were reported, showing a novel biochar application pathway for potential practical application immediately prior to manure pump-out and seasonal application to the

land. However, more testing and scaling up research for both long- & short-term biochar application to manure should be planned, and different types of biochar and effects on targeted gases should be examined.

The agronomic effects of the proposed biochar treatment of manure were recently reported by Banik et al. (2020) [60]. There is a potential for improving the sustainability of the swine and crop production systems. First, biochar can be used to mitigate emissions from manure while adsorbing nutrients from manure and, at the same time, preventing losses to the atmosphere. Banik et al., (2020) [60] reported that using biochar and manure mixture significantly increased the organic matter, total carbon (C), and total N of the soil. Therefore, we propose to use biochar treatment to improve air quality inside barns, improve worker and animal safety during manure agitation (e.g., Chen et al., (2020b) [59], and to valorize biochar-treated manure. The overall effect would improve N &C cycling in the swine-crop production systems (Banik et al., 2020) [60].

## 5. Conclusions

This laboratory-scale experiment evaluated two biochar treatments and their effective duration on mitigating targeted gases and odor emissions (as the percent reduction, % $R$) from swine manure surface in three trials. Overall, both biochars were effective in mitigating $NH_3$, $H_2S$, and odorous phenolic VOCs. The impact on GHGs, isobutyric acid, and the odor was mixed with some mitigation and generation effects. The manure and biochar type and properties had an impact on the mitigation effect and its duration.

In the case of RO treatment, there was a:

i.　significant percent reduction (% $R$) that ranged from 19–39% for $NH_3$ in all three trials, and the types of manure tested.

ii.　$H_2S$ was mitigated (16–23% reduction), but not significantly for all trials.

iii.　$CH_4$ was generated (−26 to −104%), but not significantly for all trials. $CO_2$ trended toward mitigation (0–25%), but not significantly for all trials. There was no significant effect on $N_2O$ (−1~7%).

iv.　Overall, the phenolic VOCs had relatively high % $R$ in most trials. Significant (66–78%) reductions for *p*-cresol were observed for all trials. The % $R$ for skatole ranged from 29–72%, but not significantly for all trials. The % $R$ for phenol and indole ranged from (−13–94%) and (−11–84%) but not significantly for all trials. The % $R$ for isobutyric acid ranged from (9–26%) but not significantly for all trials.

v.　Finally, the odor was mitigated in all trials (17~30%) but not significantly for all trials.

The RO treatment resulted in the decreasing effectiveness (% $R$) for $NH_3$ mitigation, consistently for the first week, the first 2 weeks, the first 3 weeks, and the entire trial for all three trials (Table 4). The % $R$ for *p*-cresol was relatively steady. However, % $R$ for several targeted gases ($H_2S$, phenol, skatole, indole) were relatively steady in at least one of the trials.

In the case of HAP treatment:

i.　the % $R$ for $NH_3$ ranged from 4–21% but not significantly for all trials.

ii.　$H_2S$ was mitigated (2–22% reduction), but not significantly for all trials.

iii.　$CH_4$ was generated (−41to −221%), but not significantly for all trials. $CO_2$ trended toward generation (−57–2%), but not significantly for all trials. There was no significant effect on $N_2O$ (−6–46%).

iv.　Overall, the phenolic VOCs had relatively high % $R$ in most trials. Significant (91–97%) reductions for *p*-cresol were observed for all trials. The % $R$ for skatole ranged from 74–95% and was significant for all trials. The % $R$ for phenol and indole ranged from (60–99%) and (29–94%) but not significantly for all trials. The % $R$ for isobutyric acid ranged from (8-88%) but not significantly for all trials.

v.    Finally, the odor was generated in all trials (−12 to −59%) but not significantly for all trials. It is important to mention that the odor measurement does not capture odor character.

The HAP treatment resulted in the decreasing effectiveness (% *R*) for $NH_3$ mitigation, consistently for the first week, the first 2 weeks, the first 3 weeks, and the entire trial for all three trials. The % *R* for *p*-cresol and skatole were relatively steady. However, % *R* for several targeted gases ($H_2S$, phenol, indole) were relatively steady in at least one of the trials. $CH_4$ generation increased with the duration of most of the trials.

This lab-scale experiment showed that treatments of biochars with different properties result in different mitigation or generation effects for the targeted gases over the duration of trials. In future studies, various types of biochar can be explored based on the desired mitigation or generation effect of targeted gases. The opportunity of biochar reapplication can also be evaluated since the greatest reduction for $NH_3$, and some phenolics were observed within the first few weeks of the study. Research with larger volumes and deeper storage of manure are needed to further advance biochar treatments.

**Supplementary Materials:** The following are available online at http://www.mdpi.com/2073-4433/11/11/1179/s1, Table S1: Review of research on uses of biochar as a manure additive and tests its effect on gaseous emissions, Table S2: Review of research on uses of biochar as a soil amendment and its effect on gaseous emissions, Table S3: Review of research on uses of biochar as an adsorbent for different gases in laboratory-scale experiments, Table S4: Biochar (top table: RO; bottom table: HAP) effect on the mitigation of $NH_3$; duration of treatment effectiveness. The mitigation effect is expressed as the % reduction (% *R*) defined as the relative difference between control & treatment (Equation (1)). Bold signifies statistical significance ($p < 0.05$). A negative value of '% R' signifies generation, Table S5: Biochar (top table: RO; bottom table: HAP) effect on the mitigation of $H_2S$; duration of treatment effectiveness. The mitigation effect is expressed as the % reduction (% *R*) defined the relative difference between control & treatment. Bold signifies statistical significance ($p < 0.05$). A negative value of '% reduction' signifies generation, Table S6: Biochar (top table: RO; bottom table: HAP) effect on the mitigation of $CH_4$; duration of treatment effectiveness. The mitigation effect is expressed as the % reduction (% *R*) defined the relative difference between control & treatment. Bold signifies statistical significance ($p < 0.05$). A negative value of '% reduction' signifies generation, Table S7: Biochar (top table: RO; bottom table: HAP) effect on the mitigation of $CO_2$; duration of treatment effectiveness. The mitigation effect is expressed as the % reduction (% *R*) defined the relative difference between control & treatment. Bold signifies statistical significance ($p < 0.05$). A negative value of '% reduction' signifies generation, Table S8: Biochar (top table: RO; bottom table: HAP) effect on the mitigation of $N_2O$; duration of treatment effectiveness. The mitigation effect is expressed as the % reduction (% R) defined the relative difference between control & treatment. Bold signifies statistical significance ($p < 0.05$). A negative value of '% reduction' signifies generation, Table S9: Biochar (top table: RO; bottom table: HAP) effect on the mitigation of phenol peak area count (PAC); duration of treatment effectiveness. The mitigation effect is expressed as the % reduction (% *R*) defined as the relative difference between control & treatment. Bold signifies statistical significance ($p < 0.05$). A negative value of '% reduction' signifies generation, Table S10: Biochar (top table: RO; bottom table: HAP) effect on the mitigation of p-cresol peak area count (PAC); duration of treatment effectiveness. The mitigation effect is expressed as the % reduction (% *R*) defined as the relative difference between control & treatment. Bold signifies statistical significance ($p < 0.05$). A negative value of '% reduction' signifies generation, Table S11: Biochar (top table: RO; bottom table: HAP) effect on the mitigation of skatole peak area count (PAC); duration of treatment effectiveness. The mitigation effect is expressed as the % reduction (% *R*) defined as the relative difference between control & treatment. Bold signifies statistical significance ($p < 0.05$). A negative value of '% reduction' signifies generation, Table S12: Biochar (top table: RO; bottom table: HAP) effect on the mitigation of indole peak area count (PAC); duration of treatment effectiveness. The mitigation effect is expressed as the % reduction (% *R*) defined as the relative difference between control & treatment. Bold signifies statistical significance ($p < 0.05$). A negative value of '% reduction' signifies generation, Table S13: Biochar (top table: RO; bottom table: HAP) effect on mitigation of isobutyric acid peak area count (PAC); duration of treatment effectiveness. The mitigation effect is expressed as the % reduction (% R) defined as the relative difference between control & treatment. Bold signifies statistical significance ($p < 0.05$). Negative value of '% reduction' signifies generation, Table S14: Biochar (top table: RO; bottom table: HAP) effect on the mitigation of odor concentration (OU/m3); duration of treatment effectiveness. The mitigation effect is expressed as the % reduction (% *R*) defined as the relative difference between control & treatment. Bold signifies statistical significance ($p < 0.05$). A negative value of '% reduction' signifies generation, Figure S1: Flow diagram of the experimental setup for the whole research. An example photo of a Trial is illustrated in Figure 1, Figure S2: Process of collecting gas samples emitted from manure for analyses of $NH_3$ and $H_2S$: (a) The manifold is attached to a block of containers (i.e., replicated treatments of the same kind); (b) Tedlar bag filled with the gas sample in the vacuum chamber; (c) The Tedlar bag is attached to the real-time analyzer, Figure S3: GHG sample collection: (a) extracting gas sample using a syringe; (b) injecting the gas sample in a vial; (c) measuring GHG concentration with a gas chromatograph, Figure S4: VOC sample collection process: (a) Flushing the glass bulb with sample air for 15 min; (b) SPME is injected in the bulb through septa to collect VOCs; (c) SPME is desorbed into a GC-MS for separation and identification of target odorous VOCs, Figure S5: HAP biochar crusted, and headspace between biochar and manure surface was

created, Figure S6: Overview photos of each treatment after the end of the trial 3: (a) HAP biochar completely sank and the manure surface crusted; (b) half of applied RO biochar became wet and floated on the manure surface; (c) some crust was observed on the surface of control manure, Figure S7: Results of FTIR analyses can explain the hydrophobicity and floating behavior when biochars are surficially applied to manure, Figure S8: Biochar effect on $NH_3$ emissions from swine manure: (a) Trial 1; (b) Trial 2; (c) Trial 3. The vertical line on Day 0 represents biochar addition. '-2 day' data signifies the pre-trial and measurement of emissions before biochar treatment. Note: emitting surface for flux estimation was assumed to be equal to $19 \times 14.5$ cm, Figure S9: Biochar effect on $H_2S$ emission from swine manure: (a) Trial 1; (b) Trial 2; (c) Trial 3. The vertical line on Day 0 represents biochar addition. '−2 day' data signifies the pre-trial and measurement of emissions before biochar treatment. Note: emitting surface for flux estimation was assumed to be equal to $19 \times 14.5$ cm, Figure S10: Biochar effect on $CH_4$ emission from swine manure: (a) Trial 1; (b) Trial 2; (c) Trial 3. The vertical line on Day 0 represents biochar addition. '−2 day' data signifies the pre-trial and measurement of emissions before biochar treatment. Note: emitting surface for flux estimation was assumed to be equal to $19 \times 14.5$ cm, Figure S11: Biochar effect on $CO_2$ emission from swine manure: (a) Trial 1; (b) Trial 2; (c) Trial 3. The vertical line on Day 0 represents biochar addition. '−2 day' data signifies the pre-trial and measurement of emissions before biochar treatment. Note: emitting surface for flux estimation was assumed to be equal to $19 \times 14.5$ cm, Figure S12: Biochar effect on $N_2O$ emission from swine manure: (a) Trial 1; (b) Trial 2; (c) Trial 3. The vertical line on Day 0 represents biochar addition. '−2 day' data signifies the pre-trial and measurement of emissions before biochar treatment. Note: emitting surface for flux estimation was assumed to be equal to $19 \times 14.5$ cm, Figure S13: Biochar effect on phenol PAC from swine manure: (a) Trial 1; (b) Trial 2; (c) Trial 3. The vertical line on Day 0 represents biochar addition. '−2 day' data signifies the pre-trial and measurement of PAC before biochar treatment, Figure S14: Biochar effect on p-cresol PAC from swine manure: (a) Trial 1; (b) Trial 2; (c) Trial 3. The vertical line on Day 0 represents biochar addition. '−2 day' data signifies the pre-trial and measurement of PAC before biochar treatment, Figure S15: Biochar effect on skatole PAC from swine manure: (a) Trial 1; (b) Trial 2; (c) Trial 3. The vertical line on Day 0 represents biochar addition. '−2 day' data signifies the pre-trial and measurement of PAC before biochar treatment, Figure S16: Biochar effect on indole PAC from swine manure: (a) Trial 1; (b) Trial 2; (c) Trial 3. The vertical line on Day 0 represents biochar addition. '−2 day' data signifies the pre-trial and measurement of PAC before biochar treatment.

**Author Contributions:** Conceptualization and supervision J.A.K., R.C.B., A.B., and Z.M.; methodology, Z.M., B.C., M.L., J.W., C.B. and S.B.; formal analysis and investigation, Z.M., and C.B.; writing—original draft preparation, Z.M.; writing—review and editing, Z.M., J.A.K., B.C., and A.B. supervision, J.A.K. and A.B.; project administration, J.A.K. and R.C.B.; funding acquisition, J.A.K. and R.C.B. All authors read and approved the final manuscript.

**Funding:** This research was partially funded by the U.S. Department of Energy—National Institute for Food and Agriculture, grant 2018-10008-28616: 'Valorization of biochar: Applications in anaerobic digestion and livestock odor control (2018–2020, PI R.B.). In addition, this research was partially supported by the Iowa Agriculture and Home Economics Experiment Station, Ames, Iowa. Project no. IOW05556 (Future Challenges in Animal Production Systems: Seeking Solutions through Focused Facilitation) sponsored by Hatch Act and State of Iowa funds. The authors would like to thank the Ministry of Education and Science of the Republic of Kazakhstan for supporting Z.M. with an M.S. study scholarship via the Bolashak Program. The authors would like to thank the Fulbright Poland Foundation for funding the project titled "Research on pollutants emission from Carbonized Refuse Derived Fuel into the environment," completed by A.B. at the Iowa State University.

**Acknowledgments:** The authors would like to thank David Laird for helpful discussions, the Iowa State University Honors Program for facilitating the research mentoring matching for Ana DiSpirito and Jordan Wright via the Honors First-Year Mentor Program.

**Conflicts of Interest:** The authors declare no conflict of interest. The funders had no role in the design of the study; in the collection, analyses, or interpretation of data; in the writing of the manuscript, or in the decision to publish the results.

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
