# Peer review of "Mitigation of Gaseous Emissions from Swine Manure with the Surficial Application of Biochars"

_atmosphere, doi:10.3390/atmos11111179_

Round 1
Reviewer 1 Report
The present manuscript conducted a 30-day lab experiment to test the effects of biochar on mitigating gas pollutants from storage swine manure. This study has investigated the interaction mechanisms between biochar and different manure and measured multiple gas pollutants to explore the biochar’s mitigation capacity. However, I think that clearer description of the experiment should be provided, and the authors should check it again that if their conclusions are supported by the data given in the tables. Also, as a preliminary study, the authors should summarize their experimental results and provide suggestions for future studies. The following are my questions and comments:
L115: Is “the assessment on…” a new bullet? Change lines?
L133: What were the ages of manure samples used in the experiment? How were they collected?
L173: This setup has limited capacity in testing the gases generated from anaerobic conditions due to the insufficient depth of the containers.
L212-213: Clarify “Gas was collected in triplicates”. Are the triplicates from the same gas bag, or collect gas continuously to three different gas bags and take samples from each of them?
L258: Insufficient description of the statistical tests. Need to list and explain the regression model.
L293: Total N content of what?
L322-L332: It says “The last statistically significant reduction was measured on day 15” here, however, it shows in Table A1 that it is still significantly reduced emission for the third week. Is it correct?
L345: If the triplicates were from the same gas bag where all gas samples were pooled together, the standard error bars do not make sense here. In this case, the differences between the measurement results are not from individual headspaces but caused by incomplete mixing in the gas bag.
L455-457: Improve the clarity of this sentence.
L564: Are these tables showing weekly mean? What is the sampling frequency for each gas? As it is a 30-d experiment, why it only shows the results from the first three weeks?
L497: I cannot see it from the data in Table A3 and A4 that the conclusion “The experiment demonstrated that biochar has the potential to mitigate ammonia, methane, carbon dioxide, and phenol…” is supported.
Reviewer 2 Report
Comments and Suggestions for Authors
The manuscript “Mitigation of gaseous emissions from swine manure with the surficial application of biochars” evaluates the effect of the surficial application of two types of biochars (i.e. highly alkaline and porous (HAP) biochar, and red oak (RO) biochar), on the mitigation of gaseous emissions from swine manure.
The overall interest of using additive at farm scale to mitigate especially ammonia and methane emissions is high.
The lab-scale experiment shows a promising approach to test the effect of manure additives for mitigating ammonia and GHG emissions from manure storage.
There is no doubt that the authors have performed a lot of work, but at this point, the manuscript lacks essential information for giving the reader the ability to understand the experimental setup, to evaluate the reliability and validity of the results of this study. Also, the conclusion claims findings not proven from the data. Therefore, I recommend reconsidering the manuscript after the authors addressed the following comments and issues.
Major comments
- The manuscript presents lab-scale experiments of gaseous emissions from swine manure and testing their mitigation by applying surficial biochars. But, it is not clear how the 12 containers (reactors) were setup using the 3 different types of manure (AG 450, ISF and PF), and applying the different treatments (i.e. RO biochar and HAP biochar). From lines 196 to 199, it is mentioned “each trial consisted of testing 4 reactors of fresh swine control manure, 4 reactors of fresh swine manure treated with RO biochar at a dosage of 1.65 kg m-2, and 4 reactors of fresh swine manure treated with HAP biochar at a dosage of 2 kg∙m-2 (Table 5)”. This means, there were 36 reactors (12 reactors x 3 trials). A diagram showing the configuration of the 36 reactors by type of manure (trial) used and treatment applied would be helpful (for reference see: Cluett Jessie et al., 2020, Animals).
- The fact that gaseous were produced using “reactors” at lab-scale, it is essential to provide information about the keys parameters influencing gas production from manure storage including the temperature of incubation and the inoculum type. This manuscript doesn’t mention these parameters. Therefore, adding these parameters would help the reader to understand more the potential of gas production in each reactor. Also, the duration of the trials (i.e. 30 days) has not been justified. Why trials were stopped after 30 days? What was the percent of daily gas produced of the total cumulative gas produced when stopping the trials?
- The procedures of gas concentration measurements are well detailed, but there are no explanations of how fluxes are calculated, as well the percent of reduction of gas emissions. Although, authors refer to Maurer et al., 2017 (line 224), it is essential to describe the procedures of flux rates calculations, and provide all data used in the calculations. This would help the reader to understand and verify the results reported. In addition, it is essential to explain how daily flux was summarized, and to calculate the cumulative emissions over the entire trials duration (i.e. 30 days) for each treatment. I recommend to use the cumulative data to test the significant differences between the treatments. A one-way ANOVA could be used to compare, for example, cumulative NH3 emissions between treatments and control.
- The values in the table 4 and 6 are reported without their standard deviations while three replicates were used for manure analysis. It is important to add the standard deviation of each value, which would help the reader to test the significant differences between the trials, as well as the treatments.
Specific comments
Introduction:
Line 55: the statement “farmers for mitigation of emissions” could be revised as most farms use manure additives for practical and agronomic purposes.
Lines 84 to 85: Table 1 is not necessary as it is not given any relevant information for the study.
Lines 86 to 93: this paragraph related to the application of biochar on soil should be removed (including table 2), as the biophysical properties of soil and manure are different. Therefore, the behavior of biochar in soil and in manure could be different.
Lines 96 to 97: I don’t think if table 3 is necessary.
Overall, the introduction needs a lot of improvements. The introduction must be more focus and transition between paragraphs smooth.
Materials and methods
Line 151: table 4, standard deviations must be added to the values, as three replicates were used for the analysis.
Line 201: Table 5, should be replaced by a comprehensible diagram showing the configuration of all 36 reactors used, and mention the total volume of manure and quantity of biochar added in each reactor.
Line 180: what type of gas was used to flash the headspace? What is the potential effect of this gas sur the gaseous measured?
Line 194: What does the stabilization consist of? What was the goal? Did you evaluate the effect of 1 week of stabilization on the gas production?
Line 225: relatively high concentrations (i.e. 10.5 and 20.5 ppm) of methane were used for the calibration. That means, the GC might not detect low concentrations of CH4, which could impact substantially the overall methane emissions measured. What was the range of CH4 concentration measured?
Lines 226 to 229: it not clear how gas sampling was done for each reactor. Are triplicates of gas samples were taken from each reactor? Or from the block of 4 same treatment?
In general, the section materials and methods must be clearly: 1. explain how the gas sampling was done? 2. Describe how the flux of each gas was calculated? 3. Describe how fluxes were summarized? And 4) how the statistical tests were performed? At this point, the manuscript is not clear on those points.
Results
Line 298: Table 6, what is the difference between pre-trial and control samples? How results in table 4 are compared to pre-trial results in table 6? STDs must be added to the tables. What is the amount of biochar added in each reactor? What is the TS content in the biochars itself? What are the impacts of the biochars on the TS content in the reactors? These points need to be include in the discussions.
Lines 300 to 313: the sinking or floating of the biochar is explained by their hydrophobicity, which depends on substrate type and the temperature. However, the temperature of the manure has not been provided in this study. This point needs to be clarified.
Lines 314 to 320: the floating or sinking of biochar is justified by the total solids content in each trial. However, it is not possible to verify if the differences between the trials are significant, as the STDs are not provided. Moreover, “line 319” seems to say that total solids total in trial 3 is higher than in trial 1. But in table 4, trial 3 has a TS content of 2.60 % and trial 1 a TS content of 2.64%. How this can be explained?
Line 345: In Figure 8 presents NH3 emissions for each trial (i.e. type of manure). What explain the negative fluxes? In addition, the comparisons between treatments are made on daily basis (lines 323 to 332), which might not be representative of the overall trend over the entire trial period (i.e. 30 days). Comparisons must be applied on the results obtained over entire study (i.e cumulative or averaged emissions). Moreover, on average (based on graphs in figure 8), hourly NH3 emissions from the untreated manure (control) over the entire study (for each trial) were ranged “approximately” between 0.04 to 0.08 g h-1 m-2 (mg converted to g). These values are at less 50% less than the baseline of hourly average of NH3 emissions from pig slurry stored in lagoons and tanks (i.e. 0.15 and 0.24 g h-1 m-2, respectively) reported in a comprehensive review of literature on “ammonia and greenhouse gas emissions from slurry storage” (Thomas et al., 2020, AEE). What explain these low NH3 emissions from the manure control? Also, what explain the “apparent” differences on NH3 emissions between the three trials? These points need to be addressed in the manuscript.
Line 372, section 3.3: CH4 fluxes presented in figure 9 seem very low. However, there is no a relevant reference, in the literature, to compare to as methane emissions from manure storages are usually scaled by the volume of manure stored (not the surface) and/or the total volatile solids (VS) stored. Thereby, to put these emissions in context, they must be scaled by the volume of manure in the reactor, as well the total VS in the container at the beginning of the trials. Again, comparisons between treatments and trials must be done in the cumulative or averaged emissions over entire trials period.
Lines 394 to 398: comparison must be done over the entire trial (i.e. on the cumulative or averaged emission). In figure 10, it doesn’t look like there a significant differences between treatments in trials 1 and 2. In trial 3, differences between treatments look evident with higher CO2 emissions for HAP treatment. What could explain the higher CO2 emissions from HAP treatment?
Globally, there is a critical lack of information to help the reader to evaluate the reliability and the quality of the results. Analyses should be done the results from over the entire trials period (i.e. results on column “Over the trial” in the tables in appendix). Figures should show the cumulative emissions scaled by surface for NH3, CO2 and N2O, and VS stored for CH4. Also, I don’t think if it is necessary to include results of the other gases, as according to the results (in appendix , column “over the trial”) there are no consistent effects of biochars treatments on the emissions of these gaseous.
Conclusion
Lines 488: “Both biochars showed the highest NH3 reduction on the day after application; then the treatments became less effective”. This claim is not valid over the entire trial duration. Only RO bicohar has shown a consistent reduction of NH3 emissions ranged between 19 to 38%.
Lines 491: it is not possible to claim a reduction of CH4 emissions, as over the entire trials, CH4 emissions increased up 200 %, which agree with Maurer et al., (2017) who reported an increase in CO2 & CH4 emissions due to the biochars.
Appendix
All graphs must show cumulative emissions.
Units of gas fluxes must to be added in the tables. Not necessary to show the intermediate (i.e. weekly) tests. Please focus discussions on the results over the entire trials.
Reviewer 3 Report
This paper describes laboratory experiments conducted using biochars to mitigate emissions from swine manures. The data collected is useful and would be of interest to researchers in the area. The overall style and content, however, need to be edited down substantially. The overall impression I have of this manuscript is that it is a masters thesis squeezed into the journal's format with very little attention to honing the story and content to clearly describe the methods and summarize the results. Much of the writing needs to be examined sentence by sentence to ensure that it directly relates to describing the issue and proposed research, clearly describes the methods without a lot of added verbiage, and concisely summarizes the findings. For instance, in the introduction there are three tables summarizing other studies of biochar. You could easily report that a series of studies have been conducted examining the impact of biochar on various emissions or soil parameters. Identify a few and give the range and major conclusions. Cite the papers but don't do an in depth analysis and cross comparison. Option 2 is that you split that out and create a separate review paper. What you did report was confusing--percentages (of what?). I assume these are reductions, but can't this be reported a bit more clearly/directly? The results of biochar in soil is extensive, why not just focus on the work in biochars in manures? Perhaps come back around in the discussion to discuss how biochar in manure is different from biochar in soil?
Materials and methods could be much clearer and not as wordy. For instance in the first paragraph on manure. Just report that three swine manure sources from Iowa were evaluated in three sequential trials. To me it is more important to know that they all came from three deep pit finishing sources. Were they all the same? Also, most analyses can be stated much more clearly. For instance determining total solids. "Total solids were determined by mass loss after drying [37]."
Description and photos of the methods--again this is something I'd expect in a thesis and not a paper. Cite the thesis if you want but don't fill the pages with extraneous information (Fig 3).
Statistics--the best method would be a repeated measures ANOVA since you are going into the pans repeatedly over time. Were interactions tested in the model? The stats presented later on are all on reductions relative to control. If that is what you ran stats on, then you need to describe that, but that is not what you'd have just be running stats on the fluxes. I think you can do a lot more on the data analysis and condense the results (Just report % reduction in a table for the three trials--plots of concentrations could all go into the supplemental).
Think of some alternate ways to summarize the data and the story you want to tell. Six tables and 11 multi-panel figures just in the body of the paper is a bit much.
I hope this review will be helpful for a future draft of the manuscript--I think the research and approach is very interesting! However, there is a lot of work that needs to be done before it's ready to be published.
Round 2
Reviewer 1 Report
The authors have made significant improvements in the description of the experiments, data presentation, discussion of the results, and conclusions. I think this manuscript is now of publishable quality.
Reviewer 2 Report
Lines 69 to 70: the abbreviations of "O, H, N and C" should be replaced by their “words” , then put abbreviations in parentheses.
Lines 123 to 124: it is highly recommended to have a least 3 replicates for each type of treatment for manure analysis (to consider in future studies).
Line 310: “at at least”. One “at” should be removed.
Method:
There is no explanations of how fluxes are calculated. From figures S8 to S12, fluxes are presented, but no explanations of how the values are calculated in “Method”. I think, it is necessary to add a session explaining how fluxes have been calculated. Moreover, the percent of reduction was calculated using the “concentrations” (Table 4), but not on the fluxes. So, what is the importance of showing the fluxes? This needs to be clearly.
Author Response
Please see the attachment. We addressed all comments and revised the manuscript.

Reviewer 3 Report
I was pleasantly surprised how quickly you made changes to the manuscript. It is much easier to read and excellent work! Thank you for taking the time to address my comments--I know they were extensive but you did great work honing the paper down to a very good manuscript.